# Dynamic Variations in Endogenous Peptides in Chinese Human Milk Across Lactation and Geographical Regions

**DOI:** 10.3390/nu17193131

**Published:** 2025-09-30

**Authors:** Baorong Chen, Kaifeng Li, Xiaodan Wang, Wenyuan Zhang, Sun Han, Yumeng Zhang, Yunna Wang, Xiaoyang Pang, Qinggang Xie, Jing Lu, Shilong Jiang, Shuwen Zhang, Jiaping Lv

**Affiliations:** 1Institute of Food Science and Technology, Chinese Academy of Agricultural Science, Beijing 100193, Chinalvjiapingcaas@126.com (J.L.); 2Feihe Research Institute, Heilongjiang Feihe Dairy Co., Ltd., Beijing 100016, China; likaifeng@feihe.com (K.L.);; 3Beijing Advanced Innovation Center for Food Nutrition and Human Health, Beijing Engineering and Technology Research Center for Food Additives, Beijing Technology and Business University, Beijing 100048, China

**Keywords:** human milk, endogenous peptides, lactation, Chinese cities, maternal diet

## Abstract

**Background/Objectives**: This study characterized the endogenous peptide profile of human milk from a Chinese multicenter cohort (*n* = 200 mothers) using the Orbitrap Fusion Lumos LC-MS/MS. **Methods**: Samples were collected across different lactation stages (2 and 6 months postpartum) and seven geographic regions (Beijing, Chengdu, Guangzhou, Jinhua, Lanzhou, Weihai, and Zhengzhou). **Results**: In total, 6960 peptides derived from 621 proteins were identified. Peptides from the polymeric immunoglobulin receptor (PIGR) were more abundant in the 2nd month than the 6th month, providing a high antimicrobial activity and immune functions for the infants. Moreover, region-specific variations were observed, with milk from Lanzhou exhibiting significantly higher levels of β-casein (CASB) and butyrophilin subfamily 1 member A1 (BTN1A1) peptides compared to other cities. **Conclusions**: Furthermore, maternal dietary intake of oils and total fat correlated positively with the intensity of specific antimicrobial peptides, including CASB_199–216, CASB_200–226, and CASB_201–226. Infant growth parameters were inversely correlated with several antimicrobial peptides, although CASB_200–225 demonstrated positive associations. These findings offer novel insights into the dynamics of endogenous peptides in human milk and may guide breastfeeding recommendations and infant formula design.

## 1. Introduction

Human milk represents the optimal source of infant nutrition, delivering essential bioactive components [1,2]. This complex fluid functions as a dynamic system enriched with proteins and proteases, which facilitate functional peptides to support infant development [3]. The majority of these peptides originate from αs1-casein, with significant contributions also derived from β-casein and αs2-casein. Endogenous peptides, constituting a fraction of the non-protein nitrogen [4], are naturally occurring peptides formed without exogenous enzymatic hydrolysis [5,6]. Their generation is primarily associated with the activity of endogenous milk peptidases, including plasmin and cathepsins D, B, and G [7]. These endogenous peptides exhibit diverse bioactive properties [8,9], encompassing antimicrobial and immunomodulatory activities [10], antihypertensive and opioid effects, promotion of intestinal wound healing, antioxidant actions, and angiotensin-converting enzyme (ACE) inhibition [11]. Notably, endogenous milk peptides, particularly those of low molecular weight, can be absorbed directly through the infant intestine [12].

Endogenous peptide levels in human milk are modulated by multiple factors, including lactation points, pH, storage conditions, delivery mode, and digestion [11]. Notably, foremilk exhibits higher levels of endogenous peptide than hindmilk, likely due to elevated aminopeptidase and carboxypeptidase activity, as well as enhanced proteolysis of β-casein, osteopontin, αs1-casein, and mucin-1 [13]. Furthermore, Dallas et al. reported increased endogenous peptide levels in milk from preterm deliveries compared with term deliveries [14]. While pH alterations did not significantly alter total peptide concentration, they modified the peptide profiles derived from specific proteins, such as β-casein, polymeric immunoglobulin receptor, osteopontin, and α-lactalbumin [3]. Regarding storage, Howland et al. demonstrated that immediate freezing of donor human milk at −20 °C or −80 °C was critical to prevent peptide degradation [11]. However, comprehensive investigations into regional variations in endogenous peptide in human milk remain limited.

Furthermore, maternal nutrition critically influences human milk quality and composition [15]. Variations in milk fat and carbohydrate content directly impact macronutrient and metabolic hormone concentrations [16], with several studies establishing links between maternal diet and overall milk composition [17,18,19]. However, current research has predominantly focused on lipidomics and oligosaccharides with maternal diet [20,21], whereas the potential influence of maternal diet on the endogenous peptide composition of human milk has been only marginally explored. In addition, China spans a vast geographic area, and maternal dietary patterns differ markedly across regions. For instance, staple food preferences vary between northern cities (e.g., Beijing and Lanzhou) and southern cities (e.g., Chengdu, Jinhua, and Guangzhou). Likewise, coastal cities (Weihai, Jinhua, and Guangzhou) are characterized by a higher consumption of seafood products compared with inland cities (e.g., Zhengzhou, Chengdu, and Lanzhou). Therefore, this study was designed as an exploratory cross-sectional analysis and analyzed differences and correlations between endogenous peptides and maternal dietary patterns in human milk samples (*n* = 200) collected different lactations (2nd and 6th months postpartum) across seven Chinese cities (Beijing, Chengdu, Weihai, Lanzhou, Jinhua, Guangzhou, and Zhengzhou).

This study aimed to: (1) compare endogenous peptide profiles in human milk collected at 2 and 6 months postpartum from mothers across seven Chinese cities representing diverse geographical regions (*n* = 200); (2) compare peptide composition among seven cities in China; (3) investigate potential correlations between peptide profiles and maternal dietary intake. These findings aim to provide a reference dataset for understanding potential lactation stages and geographic influences on milk peptides, which may inform breastfeeding guidance and infant formula design.

## 2. Methods and Materials

### 2.1. Chemicals and Samples

Acetonitrile (ACN) and formic acid (FA) were sourced from Thermo Fisher Scientific (USA), and trichloroacetic acid (TCA) was obtained from Macklin (Shanghai Macklin Biochemical Co., Ltd., Shanghai, China). This study (*n* = 200 mothers) was conducted as part of the Chinese Human Milk Project (CHMP, NCT03675204), a multicenter, cross-sectional investigation designed to evaluate human milk composition in the Chinese population [22]. Human milk samples were provided by the Chinese Human Milk Project (the CHMP study, NCT03675204) collected from mothers in seven Chinese cities: The CHMP recruited 1800 healthy lactating mothers from Beijing, Chengdu (Sichuan province), Guangzhou (Guangdong province), Jinhua (Zhejiang province), Lanzhou (Gansu province), Weihai (Shandong province), and Zhengzhou (Henan province). For the present analysis, 200 participants were randomly selected from the CHMP cohort. Human milk samples were collected between 15 and 180 days postpartum along with corresponding maternal and infant information. This study was conducted according to the guidelines of the Declaration of Helsinki, and approved by Institutional Review Board of Shanghai Nutrition Society Ethic Committee (Ethic Approval [2016] No. 006) (protocol code: 17-SM-9-FEIHE-001, date of approval: 28 September 2016). All participants provided written informed consent. Inclusion criteria were as follows: lactating women aged 20–40 years, currently breastfeeding their infants, self-reported good health, non-smoker, and non-drinker. Among participants, 70 reported exclusive breastfeeding, 28 reported mixed feeding, and 101 did not provide feeding information.

Samples were primarily collected at months 2 and 6 postpartum, transported under dry ice conditions, and stored at −80 °C in ultralow-temperature freezers until processing. Consistent with prior evidence demonstrating > 96% peptide profile reproducibility between boiled and unheated samples, protease inhibitors were omitted during sample preparation [23].

### 2.2. Dietary Frequency

Maternal dietary intake during the month preceding milk sampling was assessed using a food frequency questionnaire (FFQ). This instrument documented: (i) maternal characteristics (occupation, body weight); (ii) lifestyle factors (smoking, alcohol consumption, sleep patterns); (iii) infant health status; and (iv) quantitative food consumption. Food consumption data quantified average daily intake (g/day) over a one-week period for: flours; fruits; vegetables; and livestock products. Comprehensive characteristics of the human milk samples are presented in Table 1.

### 2.3. Sample Preparation

The milk fat fractionation was performed in the way reported by Dallas et al. [23]. Briefly, 200 μL aliquots were centrifuged at 16000× *g* for 10 min at 4 °C. Skim milk was obtained from beneath the fat layer, with centrifugation repeated until complete fat removal. Proteins were precipitated using TCA, as described by Ferranti et al. [24]: 200 μL of 200 g/L TCA in ultra-pure water was added to skim milk, mixed, and centrifuged at 3000× *g* for 10 min at 4 °C. The peptide-enriched supernatant was desalted via C18 solid-phase extraction (Waters, Milford, MA, USA). Peptides were eluted using an 80% ACN/0.1% TFA, dried under vacuum, and reconstituted for MS analysis. Endogenous peptide concentrations were quantified at 205 nm using a Thermo Scientific Nanodrop spectrophotometer (Waltham, MA, USA).

### 2.4. Mass Spectrometry Analysis

Peptide separation was performed using a Thermo Scientific Easy-nLC 1200 system (Waltham, MA, USA) with an elution gradient adapted from Dingess et al. [4]. The gradient consisted of a 50 min linear increase from 5% to 30% solvent B (0.1% FA in ACN), followed by a 3 min ramp to 50% solvent B. Mass spectrometry was conducted on a Thermo Scientific Orbitrap Fusion Lumos (USA) platform with electrospray ionization (2.4 kV). Full MS scans (400–1500 m/z) were acquired at 120 K resolution. Collision-induced dissociation was selected as the fragmentation option and the collision energy was set at 35%. The MS cycle time was 3 s with data-dependent analysis and automated precursor peak selection. Precursor ions were excluded after one fragmentation for 60 s and within 10 ppm mass error range. The following parameters were used to choose precursors for fragmentation: most intense peaks, ion-intensity threshold 5.0 × 10^3^ and charge state 2–7. The ion trap with an automated scan range was used to identify fragments. The quality control (QC) standard (293 T cell line lysate) was used to ensure analytical accuracy and reproducibility (coefficient of variation < 5% for peak intensity).

Thermo Proteome Discoverer (v 2.4) was used to explore the human milk protein sequence library for spectra. Exemplificative workflow for the identification of the peptide ETIESLSSSEESITEYK (CASB_16–33) in the 2-month breast milk sample was shown in Figure 1. And the other examples of different lactation stages and cities (e.g., Lanzhou and Jinhua) were shown in Appendix A. The human protein database was downloaded from UniProt on 18-09-2021 and used to search with peptide lengths ranging from 4 to 144. Phosphorylation of serine and threonine and oxidation of methionine were the potential modifications. Proteome Discoverer percolator was employed to distinguish between valid and wrong spectrum identifications using a decoy database search. Based on this, only peptides with a high confidence (*q*-value < 0.01) were included in the results (FDR < 1%). Peptide sequences with multiple modifications were grouped into a single peptide for counts and abundance. Counts measured the number of unique peptide sequences identified in a sample. The area under the curve of the eluted peak was used to calculate abundance (ion intensity), and log10 (intensity) was used for representation.

### 2.5. Bioactive Peptide Prediction

The Milk Bioactive Peptide Database (MBPDB, http://mbpdb.nws.oregonstate.edu/, accessed on 20 October 2022) was used to detect bioactive peptides in human milk samples [25]. The sequence search type was chosen to retrieve bioactive peptides. The similarity threshold was set to 80% and the scoring matrix was set to identity. ‘Get extra output’ was selected in order to obtain a specific percentage similarity between the query sequence and the bioactive peptide sequence.

### 2.6. Data Analysis

Peptide counts and intensities across human milk samples were compared using non-parametric tests in IBM SPSS Statistics 26 (Chicago, IL, USA). Pairwise lactation stage differences (2 vs. 6 months) were assessed with Mann–Whitney tests. Regional variations among seven cities were evaluated through Kruskal–Wallis and Wilcoxon tests, applying Bonferroni correction for multiple testing (*p* < 0.05). Data were reported as mean ± SD. GraphPad Prism 8 was used to perform the box plots, and different letters indicated significant differences (*p* < 0.05). Complex visualizations (Upset, Venn, and heatmaps) were produced using the Tutools online platform (http://cloudtutus.com:8000/product/, accessed on 7 October 2022). A correlation heatmap was constructed using Spearman’s rank correlation to assess associations among variables, with statistical significance set at *p* < 0.05.

## 3. Results and Discussion

### 3.1. Comprehensive Profiling of Endogenous Peptides in Human Milk

A comprehensive peptidomic analysis of endogenous peptides identified 6960 peptides across 200 human milk samples. The mean peptide count was 606.64 ± 156.60, with an average intensity was 2.10 × 10^10^ ± 1.39 × 10^10^, and a mean peptide concentration of 0.41 ± 0.26 mg/mL (ranging from 0.09 to 2.71 mg/mL). Notably, the number of endogenous peptides detected in this study exceeded that reported by Gan et al. (5847 peptides from 19 human milk samples) [3]. This discrepancy may be attributed to the greater sample size and diversity of human milk in the current analysis. Moreover, peptides were identified from 621 proteins, with the majority originating from β-casein (CASB, 20.33%), osteopontin (OSTP, 8.52%), polymeric immunoglobulin receptor (PIGR, 7.10%), αs1-casein (CASA1, 4.57%), and butyrophilin subfamily 1 memberA1 (BT1A1, 3.66%). These findings aligned with the study by Gan et al. [3]. In human milk samples, CASB peptides were the most abundant, followed by peptides derived from PIGR and OSTP [26,27]. Notably, peptides were predominantly localized to the CASB_16–33 and CASB_200–226 residue regions. Among them, the peptide ETIESLSSSEESITEYK exhibited the highest intensity. The peptides RETIESLSSSEESITEYK, RETIESLSSSEESITEYKQKVE, and TIESLSSSEESITEYK were also released from the CASB 16–33 fragment, all of which are known to stimulate cellular proliferation [28].

### 3.2. Dynamics of Milk Peptide Profiles Across Lactation Stages

In this study, peptide concentrations in human milk were higher at the 2nd month (0.44 ± 0.33 mg/mL) compared with the 6th month (0.37 ± 0.15 mg/mL) (*p* > 0.05). However, most previous investigations have primarily focused on protein concentrations rather than peptides. Protein constituted approximately 1% of human milk, with concentrations peaking at 14–16 mg/mL during early lactation and gradually declining to 8–10 mg/mL by 3–4 months. By 6 months, protein levels further decreased to 7–8 mg/mL [29]. This progressive decline in protein content over the first six months of lactation suggests that duration of breastfeeding may influence endogenous peptide levels [30]. Furthermore, consistent with this trend, peptide concentrations also decrease as lactation progressed.

The total counts of peptide species in human milk at the 2nd month of lactation (5652 peptides) were higher than that in the 6th-month milk (4684 peptides), indicating a progressive decline in peptide counts with the lactation prolongation [4]. This reduction may be associated with changes in both plasmin sensitivity and overall protein composition of human milk [31]. Notably, plasmin activity appeared to be influenced by postnatal time and cohort variation, with activity levels decreasing with the duration of lactation [32]. The higher peptide counts observed in the 2nd-month milk could be attributed to elevated plasmin activity during early lactation. However, no significant differences were observed in the total peptide counts or intensity across different stages of lactation (Figure 2A,B, *p* > 0.05), consistent with findings by Nielsen et al. [13]. When comparing peptide counts released by specific proteins, the 2nd-month milk released fewer peptides from CASB, OSTP, and BT1A1 than the 6th-month milk, whereas PIGR peptides were more abundant in the 2nd-month milk (Figure 2A, *p* < 0.01). Regarding protein intensity, a significant difference was only observed for BT1A1 peptides, with levels being lower in the 2nd-month milk than in the 6th-month milk (Figure 2B).

Differential peptide analysis across lactation stages was shown in Figure 2C,D. In the 2nd month human milk, the main differences in protein intensity were observed in lactoperoxidase (down-regulation), BT1A1 (down-regulation), and PIGR (up-regulation) (Appendix A). Differences in peptide intensity were primarily associated with PIGR peptides (37), CASB peptides (17), BT1A1 peptides (11), OSTP peptides (8), CASA1 peptides (7), and bile salt-activated lipase peptides (7) (Appendix A). The peptide mapping of these proteins was presented in Figure 2E.

PIGR is a transmembrane protein responsible for transporting IgA and IgM from mammary epithelial cells into milk [33]. In this study, only four peptides (PIGR_600–615, PIGR_600–617, PIGR_611–643, PIGR_625–643) were down-regulated, whereas the remaining 33 peptides were up-regulated in the 2nd-month milk. Most differential peptides were localized to the PIGR_604–646 region. Moreover, the higher counts and intensities of PIGR peptides were observed in the 2nd month human milk than in the 6th month (Figure 2A,B). This phenomenon may reflect structural protection of PIGR within the mammary gland, preserving its antimicrobial and immune functions for the infant during early lactation [14,34]. Over time, the abundance of PIGR peptides declined with lactation.

In the mammary gland, CASB was hydrolyzed by proteases including plasmin, trypsin, cathepsin D, and elastase. Due to its relatively loose structure, CASB readily interacts with plasmin at the interface of casein micelles, which facilitates the hydrolysis of casein-derived peptides [4]. The majority CASB peptides originated from its *N*- and *C*-terminal structural domains, with fewer from the central region [5], as showed in Figure 2E. In human milk, differential CASB peptides detected in the 6th month of lactation were up-regulated compared with those in the 2nd month, with peptide lengths largely ranging from 16 to 54 residues. This up-regulation may be associated with the progressive increase in CASB concentration during lactation, rising from approximately 20% in early milk to 45% in later stages. Beyond its structural role, CASB contributed to the transport of essential minerals, such as calcium and phosphorus, which were critical for metabolic processes. However, the intrinsic complexity of casein micellar structure makes digestion more challenging. This may explain the higher concentration of casein in late-stage human milk, which coincides with the period when neonatal organs and tissues have undergone sufficient structural development to better process these proteins [35].

### 3.3. Geographic Variation in Human Milk Peptide Composition

Peptide concentrations varied significantly across different cities (Table 1). Among them, Lanzhou and Weihai exhibited higher peptides content compared with other cities (*p* < 0.05). In contrast, milk samples from Jinhua (2584) contained the fewest peptides, whereas those from Lanzhou (3540) contained the most (Figure 3B). On average, 650 ± 147, 610 ± 178, 653 ± 190, 561 ± 110, 603 ± 149, 574 ± 162, and 598 ± 139 peptides were identified in samples from Beijing, Chengdu, Guangzhou, Jinhua, Lanzhou, Weihai, and Zhengzhou (*p* > 0.05), respectively (Figure 4A). There was a significant variation in protein counts between Beijing and Lanzhou, as well as Lanzhou and Zhengzhou (*p* < 0.01). The protein counts of Chengdu, Guangzhou and Jinhua were similar. Notably, the intensity of peptide derived from CASB, CASA1 and BT1A1 differed significantly among cities (Figure 4B). Furthermore, the average intensity of peptides in Lanzhou was markedly higher than that in Beijing (*p* < 0.05) and Zhengzhou (*p* < 0.01).

These peptide maps of CASB, CASA1, and BT1A1 were generated to analyze regional differences (Figure 4E). Compared with other cities, human milk from Lanzhou and Guangzhou contained higher levels of CASB peptides. The most pronounced differences in CASB peptides were observed in the range of 20–40 residues and 200–220 residues. For CASA1, the main variations occurred within residues 25–50, with Jinhua samples exhibiting the highest peptide intensity, followed by Lanzhou. This may be related to the high intake of soybeans (r = +0.46) in Jinhua (98.28 g/day) and Lanzhou (63.95 g/day) (Figure 4C,D).

Significant differences in peptide counts of BT1A1 were also observed among several cities, including Beijing and Guangzhou, Guangzhou and Jinhua, Jinhua and Weihai (*p* < 0.05). Notably, Guangzhou displayed the lowest counts of BT1A1 peptides. As BT1A1 is a primary component of the milk fat globule, influencing both lipid droplet formation and phospholipid composition in mammary epithelial cells [36,37]. Previous studies have further demonstrated that regional differences in dietary patterns strongly affect the triglyceride composition of human milk, with Guangzhou showing particularly distinct profiles [38]. Consistent with this, significant differences in BT1A1 peptide intensity were observed between Guangzhou and Jinhua, Guangzhou and Lanzhou, Jinhua and Weihai, and Lanzhou and Weihai (*p* < 0.01). Lanzhou had a higher BT1A1 intensity than other cities and may be associated with high intake of potatoes (r = +0.64) and nuts (r = +0.79) in Lanzhou (Figure 4D).

These regional variations in peptide composition may be attributed to dietary patterns associated with geographic location. For instance, diets in Weihai, Jinhua and Guangzhou were characterized by a higher intake of freshwater products [39]. In contrast, residents of Chengdu, Zhengzhou and Lanzhou, located in central and western China, tended to consume wheat-based staple foods. Meanwhile, individuals in Beijing, situated in northern China, traditionally prioritized rice and animal meat. Such differences in maternal diet [40], along with lifestyle factors and delivery mode, were likely key determinants of human milk composition [41,42,43]. Therefore, further studies are needed to investigate the impact of dietary diversity across regions and countries on peptide profiles in human milk.

### 3.4. Bioactive Peptides in Human Milk and Functional Potential

The Milk Bioactive Peptide database (MBPDB) was used to predict the functional peptides in human milk, with the findings reported in Appendix A. A total of 197 bioactive peptides were identified, including antimicrobial peptides (106), ACE-inhibitory peptides (46), and stimulates proliferation peptides (43). Most of these peptides were deprived from CASB and κ-casein, consistent with the findings of Dallas et al. [23], who reported that 99% of bioactive peptides originated from CASB.

Antimicrobial peptides play important roles in modulating immune activity and participating innate immune defense [44]. Several have been previously reported to exhibit bioactive functions. For example, CASB_197–213 peptide inhibited the growth of *Escherichia coli* and *Yersinia enterococcus* [45], while CASB_211–225 was more abundant in preterm milk compared with term milk [46]. As Figure 5A shows, the content of antimicrobial peptides was higher in human milk at the 2nd month compared with the 6th month (*p* < 0.05). This finding was consistent with the results of Baricelli et al., who reported that antimicrobial peptides were most abundant in colostrum and subsequently decreased as lactation progresses [47].

At the early stage of life, the neonatal intestinal microbiota is relatively simple and fragile, characterized by low species diversity and high susceptible to external factors. However, over time, the gut microbiota becomes relatively stable [48]. Antimicrobial peptides in human milk that it may contain bacteriostatic or bactericidal activity against newborn pathogens in vitro. Beyond direct antimicrobial activity, these antimicrobial molecules may contribute to protective mechanisms through controlling bacterial load in the gastrointestinal tract, modulating immune responses, and shaping the gut microbiome [49,50]. Thus, the dynamic intensity of antimicrobial peptides in human milk reflects the developmental needs of the newborn, supporting intestinal homeostasis and promoting healthy infant growth.

### 3.5. Associations Between Maternal Diet and Antimicrobial Peptides

Weihai exhibited the highest concentrations of antimicrobial peptides among seven cities, followed by Guangzhou. Significant differences in antimicrobial peptide intensity were observed between Weihai and several other cities, including Beijing (*p* = 0.005), Zhengzhou (*p* = 0.015), and Jinhua (*p* = 0.039) (Figure 5B). Maternal diet appears to influence both the fat composition and the immunoregulatory properties of human milk [43], a pattern that has also been reported in studies of Kuwaiti mothers [41,51]. Correlation analysis between antimicrobial peptides and maternal diet was presented in Figure 6A. Notably, both oil consumption and total fat intake were positively correlated with the intensity of several antimicrobial peptides, including CASB_199–216, CASB_200–226, and CASB_201–226. Previous studies have demonstrated that the CASB_200–226 peptide possesses antibacterial activity [52].

The activity of antimicrobial peptide has been attributed to membrane decomposition mechanisms that directly interfere with the integrity of bacterial cell membranes and cell walls [53]. Oils represent one of the most common dietary sources of fatty acids, which play important roles in enhancing intestinal cell membrane permeability and contributing to antimicrobial effects [54]. Moreover, fatty acids and vitamin D have been shown to modulate the interactions between cell membranes and antimicrobial peptides [55,56]. Several studies have suggested that structural modifications of fatty acids may provide a simple and effective strategy to enhance the activity of antimicrobial peptides [57]. However, animal studies indicated dose-dependent effects: mice fed low to moderate doses of antimicrobial peptides exhibited higher levels of short-chain fatty acid, whereas high doses increased the risk of intestinal permeability and microbial imbalance [58]. These findings showed the importance of maintaining an optimal intake of oils and fats during pregnancy, both to support maternal nutrition and to enhance the antimicrobial properties of human milk.

In contrast, dairy consumption was negatively correlated with the intensity of multiple antimicrobial peptides (CASB_196–226 and CASB_204–226, *p* < 0.05). Many factors are known to induce antimicrobial peptide expression, including albumin, arginine, butyric acid, calcium, isoleucine, vitamin D, and zinc [59]. Although dairy products are an important source of calcium [60], a high dairy intake may also be associated with endogenous metabolic effects that influence weight regulation and health [61].

Egg proteins were also recognized as potential precursors for bioactive peptide with diverse peptide sequences [62] and demonstrated angiotensin-converting enzyme (ACE) inhibitory and anti-inflammatory properties [63]. However, these previously reported effects contrast with the present findings. Palmer et al. observed a dose–response relationship between ovalbumin levels in human milk and controlled egg intake within 8 h [64]. Therefore, the further research is necessary to confirm the association between maternal egg consumption and antimicrobial peptides in human milk.

On the other hand, infant weight, length and head circumference mainly were generally negatively associated with several antimicrobial peptides (CASB_199–226, CASB_197–226, and CASB_196–225, *p* < 0.05), whereas the CASB_200–225 peptide showed a positive correlation with growth parameters (Figure 6B). Antimicrobial peptides are known to modulate immune activity and contribute to self-regulatory immune processes [44]. Antimicrobial peptides were mainly used to protect newborns from inflammation and infectious diseases, and used to fight serious infections in infancy, especially in infants born preterm or with low birth weight [65]. This protective function may explain why antimicrobial peptides appear to have a more limited contribution to direct infant growth.

This multicenter, cross-sectional study, which collected milk samples from diverse geographic regions in China, could provide a dataset milk of peptide composition. Nevertheless, several limitations were existed in our study. Firstly, the limitation of our study is the cross-sectional study design, which only reveals correlation rather than causality. Secondly, although numerous peptides with potential antimicrobial function were identified, their biological roles require further experimental validation to clarify their activity and correlation with maternal diet. Thirdly, the lack of biomarker analyses limited our ability to establish causal relationships among maternal diet, peptide profiles, and infant growth [66]. Therefore, future studies should (i) integrate longitudinal cohort designs and (ii) multi-omics biomarker analyses (including maternal blood, urine, feces, and infant blood) to more comprehensively elucidate the dietary and immune mechanisms underlying antimicrobial activity in human milk.

## 4. Conclusions

The study was the first to investigate the characteristics of endogenous peptides in human milk among different lactations and regions from Chinese women, and explore the correlation with the maternal diet. Endogenous peptides, as an important non-protein nitrogen fraction in human milk, exhibit diverse bioactive functions that support to infant nutrition and immune protection. In this multicenter study, peptide profiles variations across lactation stages and geographic regions were observed. Antimicrobial peptide content was higher in human milk at the 2nd month compared to the 6th month, suggesting a potentially temporal regulation that may align with developmental needs of the infant. Among the geographic variation in peptide composition, CASB and BT1A1 peptides were enriched in milk samples from Lanzhou. Furthermore, maternal dietary fat and oil intake showed positive associations with specific antimicrobial peptides, suggesting a potential link between maternal diet and milk peptide composition. While these findings provide valuable preliminary insights, the relatively limited sample size and restricted geographic coverage mean that the results should be interpreted with caution. Future research involving larger, longitudinal, and multi-omics cohort studies will be essential to validate these associations and to clarify the functional significance of endogenous peptides in infant development.

## Figures and Tables

**Figure 1 nutrients-17-03131-f001:**
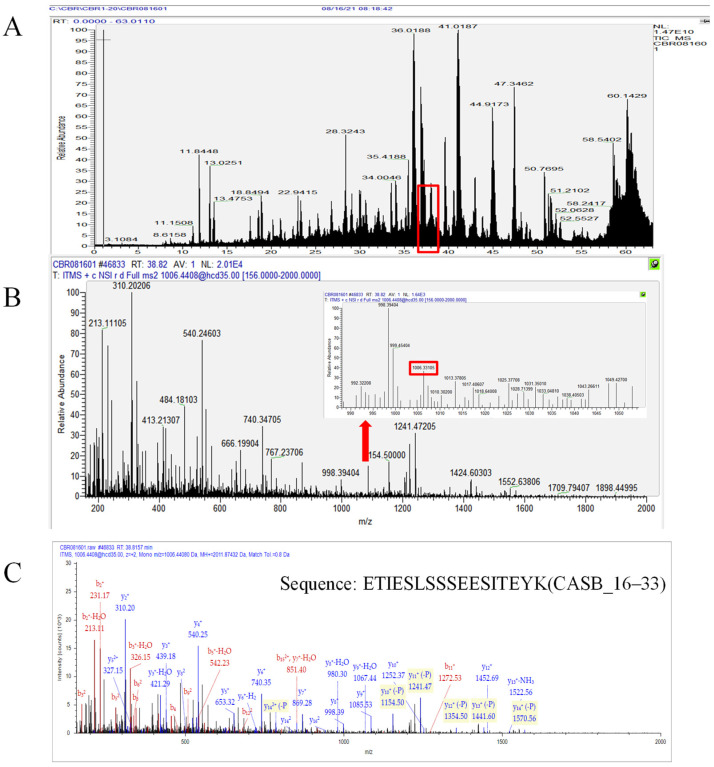
Exemplificative workflow for the identification of the peptide ETIESLSSSEESITEYK (CASB_16–33) in the 2-month breast milk sample. (**A**) HPLC-MS/MS total ion chromatogram (TIC) of endogenous peptides detected in 2-month milk. (**B**) MS 1 spectrum corresponding to the retention time (RT: 38.82 min) of peptide CASB_16–33. The inset in panel B shows a magnified view containing the m/z 1006.33 signal. (**C**) Sequence assignment of CASB_16–33 obtained from raw data analysis using Proteome Discoverer software (version 2.4, Thermo Scientific).

**Figure 2 nutrients-17-03131-f002:**
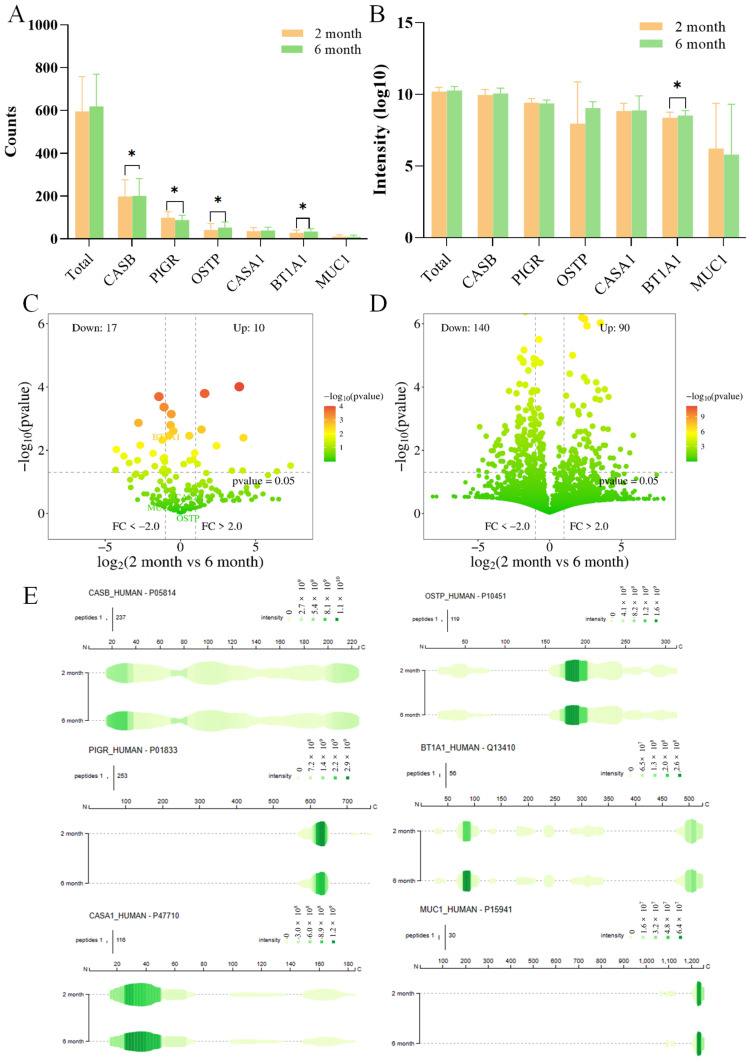
The comparison of peptide profile in different lactations. Total counts (**A**) and abundance (**B**) of peptides in lactation human milk samples. The volcano map analysis of different proteins (**C**) and peptides (**D**) in different lactation, and the peptide map of the different proteins (**E**). * indicates significantly different values among columns (*p* < 0.05). CASB, β-casein; PIGR, polymeric immunoglobulin receptor; OSTP, osteopontin; BT1A1, butyrophilin subfamily 1 memberA1; CASA1, αs1-casein; MUC1, mucin-1.

**Figure 3 nutrients-17-03131-f003:**
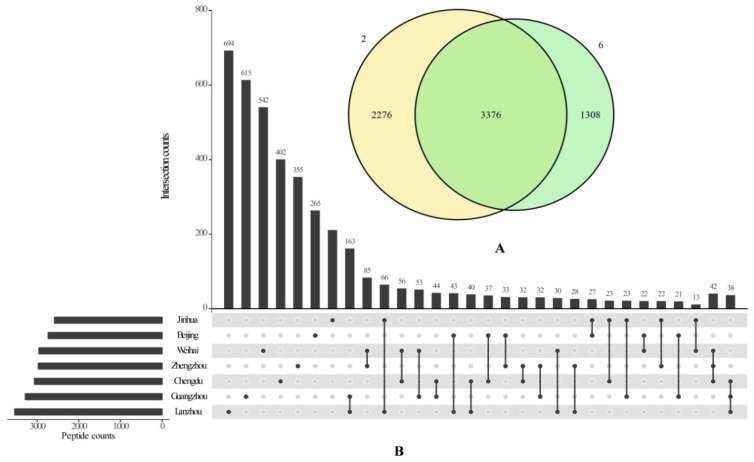
Counts of identified peptides in human milk samples and overlaps with other sample groups. (**A**) Venn diagram of number of peptides identified in human milk samples of 2nd and 6th months; (**B**) Upset diagram of number of peptides released from seven cities. In the Upset diagram, the upper bar represents the overlaps number of peptides from different cities; the bar on the left side of the figure represents the total number of peptides in every city; the lattice represents the intersection of seven cities.

**Figure 4 nutrients-17-03131-f004:**
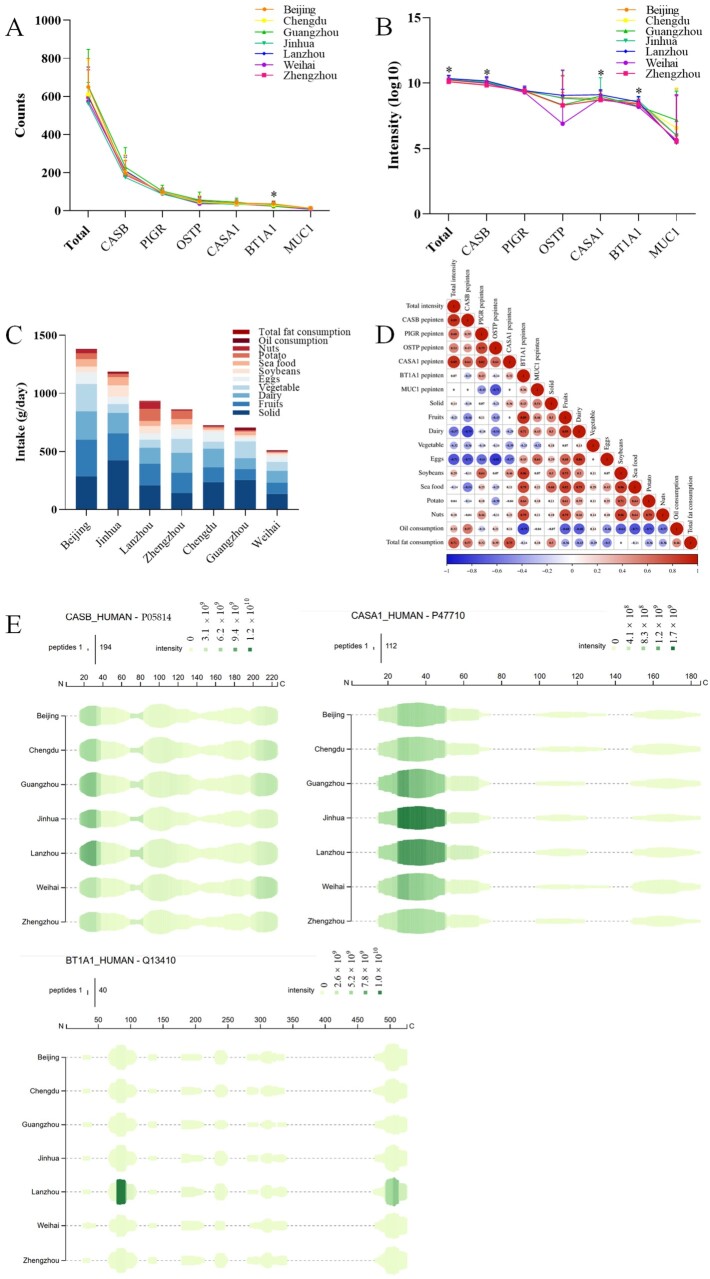
The comparison of peptide profiles in different cities. Total counts (**A**) and intensity (**B**) of peptides in seven cities human milk samples. Dietary intake in each city (**C**), the correlation analysis between peptides and diet (**D**). The peptide profile of the main protein (**E**), including β-casein (CASB), αs1-casein (CASA1), butyrophilin subfamily 1 memberA1 (BT1A1). Different letters indicate significantly different values (*p* < 0.05). * Indicates significantly different values among columns (*p* < 0.05). CASB, β-casein; PIGR, polymeric immunoglobulin receptor; OSTP, osteopontin; BT1A1, butyrophilin subfamily 1 memberA1; CASA1, αs1-casein; MUC1, mucin-1.

**Figure 5 nutrients-17-03131-f005:**
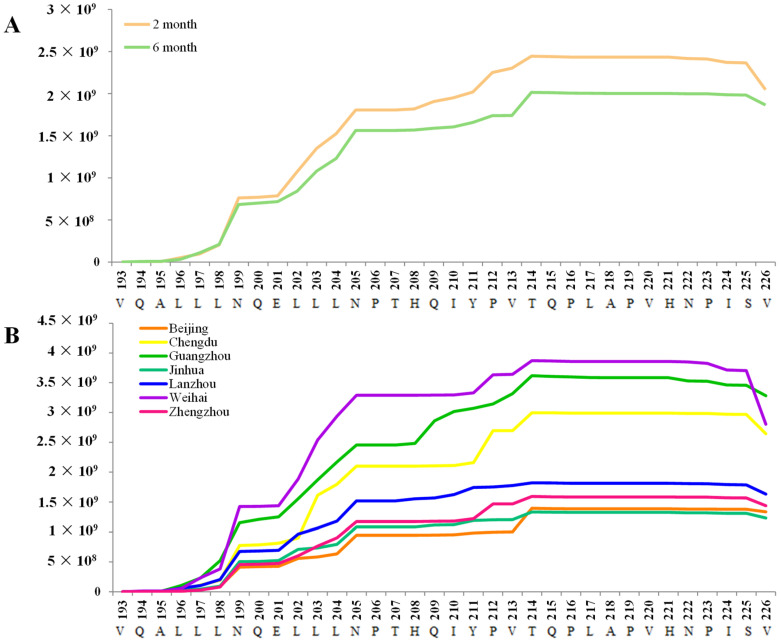
Intensity of antimicrobial peptides identified in human milk mapped on the sequence of CASB (193–226 residues) in different lactations (**A**) and cities (**B**). CASB: β-casein.

**Figure 6 nutrients-17-03131-f006:**
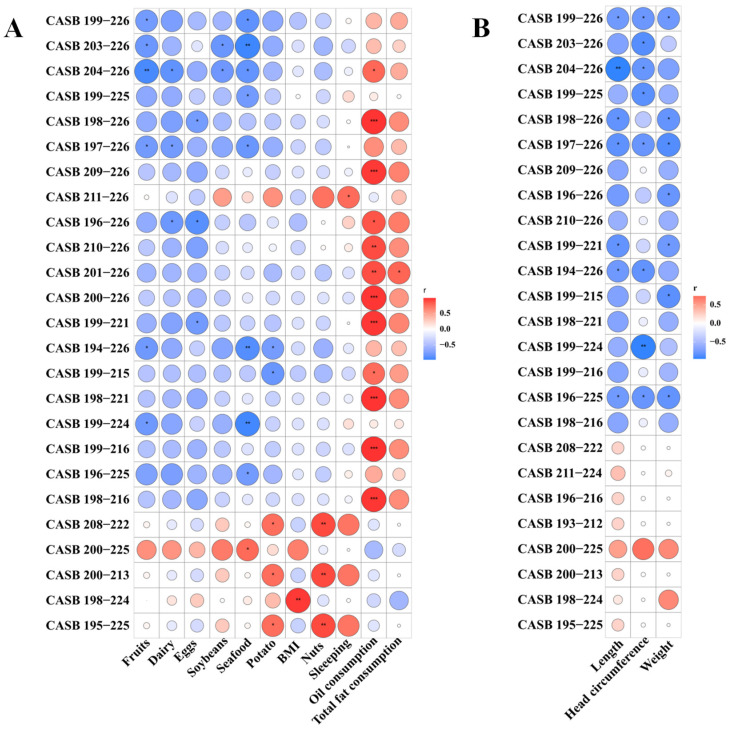
The correlations of antimicrobial peptide profiles with maternal diet (**A**) and infant growth (**B**) in different cities. *: *p* < 0.05. **: *p* < 0.01. ***: *p* < 0.001.

**Table 1 nutrients-17-03131-t001:** Demographics and sample characteristics.

Characteristics		Total	2nd Month	6th Month	Beijing	Chengdu	Weihai	Lanzhou	Jinhua	Zheng Zhou	Guang Zhou
Sample information	Number	200	106	94	28	29	20	37	30	32	24
	Peptide concentrations mg/mL	0.41	0.44	0.37	0.35	0.37	0.59	0.48	0.34	0.39	0.38
Infant characteristic	Girls (Boys)	86 (114)	46 (60)	40 (54)	14 (14)	10 (19)	12 (8)	14 (23)	12 (18)	14 (18)	10 (14)
	Length (cm)	60.83	55.79	66.50	64.57	60.01	58.06	61.97	61.49	61.34	56.48
	Weight (kg)	7.01	5.50	8.73	7.38	7.39	5.85	6.89	7.06	8.08	5.79
	Head circumference (cm)	40.50	38.34	42.96	41.70	39.57	38.37	40.25	42.20	40.28	40.02
Delivery mode	Virginal	123	67	56	18	14	16	29	19	17	10
	Cesarean	61	34	27	6	13	3	7	10	13	9
	Assisted	15	5	10	4	2	1	1	0	2	5
	other	1	0	1	0	0	0	0	0	0	0
Maternal BMI	Normal (BMI 18–24.9)	156	86	70	22	27	15	32	20	20	20
	Overweight (BMI 25–29.9)	42	20	24	6	2	5	5	9	11	4
	Obese (BMI ≥ 30)	2	0	0	0	0	0	0	1	1	0

## Data Availability

The original contributions presented in the study are included in the article/Appendix A, further inquiries can be directed to the corresponding authors.

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
