# Peer review of "Dynamic Variations in Endogenous Peptides in Chinese Human Milk Across Lactation and Geographical Regions"

_nutrients, 2025, doi:10.3390/nu17193131_

Round 1

Reviewer 1 Report

Comments and Suggestions for Authors

nutrients-3881057_Dynamic Variations of Endogenous Peptides in Chinese Human Milk Across Lactation and Geographical Regions

General assessment
The article entitled “nutrients-3881057_Dynamic Variations of Endogenous Peptides in Chinese Human Milk Across Lactation and Geographical Regions” has been submitted to the section “Pediatric Nutrition” of the special issue “Explore the Nutritional Composition of Human Milk and Its Impact on Infant Formula Development and Human Health”. This study provides a comprehensive characterization of the human milk peptidome from a multicenter Chinese cohort, identifying 6,960 peptides across lactation stages and regions. The results highlight temporal changes, such as higher PIGR-derived peptides at 2 months, as well as regional differences, including elevated β-casein and BTN1A1 peptides in Lanzhou. Associations with maternal diet and infant growth further underscore the functional relevance of milk peptides. These findings offer valuable insights with potential applications in breastfeeding guidance and infant formula design.

Abstract
At line 16, the sample size is reported as n = 200. The unit should be clarified (e.g., 200 mothers). Since the title mentions different regions, the abstract should also specify which regions were included in the study.

Introduction
The introduction emphasizes the importance of dynamic variations in endogenous peptides in Chinese human milk across lactation. However, it does not justify why variations might occur across the seven cities studied. It would be useful to clarify whether this was a nutritional hypothesis, as later discussed in the results and discussion, or whether the study should be framed as an exploratory analysis. This section should be expanded to present the scientific rationale more clearly.
The stated objectives also require revision. Given that seven cities and 200 samples are included, proportional sampling representative of each city’s population should ideally have been conducted, and the sample size might need to be larger considering the populations of these cities. The objectives should be revised and clearly stated at the end of the introduction.

Materials and Methods
The study design should be specified. It is also important to state what type of sampling was performed and whether a sample size calculation was conducted to ensure representativeness for each city. The inclusion criteria for participating mothers should be described in detail to confirm comparability across groups. In particular, it should be clarified whether all participating mothers were exclusively breastfeeding, or if any were supplementing with formula, as this could significantly influence the results.
In the statistical analysis section, it should be indicated how normality of the data distribution was assessed.

Results and Discussion
At line 172, peptide concentrations are reported at the second and sixth months, but it is not stated whether the differences are statistically significant. The same issue arises in lines 180–181. Even if this information is shown in the figures, statistical significance should be explicitly mentioned in the text. Figure 3 contains valuable information but is difficult to read due to low resolution; the quality should be improved.
The discussion would be strengthened by explicitly addressing the strengths and limitations of the study, as well as by outlining directions for future research.

Conclusion
The conclusion should be revised in light of the representativeness of the sample studied, to ensure that the claims are proportional to the data presented.

Author Response

Point 1: General assessment: The article entitled “nutrients-3881057_Dynamic Variations of Endogenous Peptides in Chinese Human Milk Across Lactation and Geographical Regions” has been submitted to the section “Pediatric Nutrition” of the special issue “Explore the Nutritional Composition of Human Milk and Its Impact on Infant Formula Development and Human Health”. This study provides a comprehensive characterization of the human milk peptidome from a multicenter Chinese cohort, identifying 6,960 peptides across lactation stages and regions. The results highlight temporal changes, such as higher PIGR-derived peptides at 2 months, as well as regional differences, including elevated β-casein and BTN1A1 peptides in Lanzhou. Associations with maternal diet and infant growth further underscore the functional relevance of milk peptides. These findings offer valuable insights with potential applications in breastfeeding guidance and infant formula design.

Response 1: We are deeply grateful for your positive assessment of our manuscript. Your recognition of our work as a valuable supplement to the existing knowledge and your commendation of the clarity and accessibility of our writing are immensely encouraging. Thank you once again for your support and encouragement. Thank you for your valuable comments.

Point 2: Abstract: At line 16, the sample size is reported as n = 200. The unit should be clarified (e.g., 200 mothers). Since the title mentions different regions, the abstract should also specify which regions were included in the study.

Response 2: Thank you for your valuable feedback. We have revised the abstract accordingly: (1) The sample size is now clearly indicated as “n = 200 mothers”. (2) The seven cities included in the study have now been explicitly listed, as shown in Lines 18-19. We appreciate your guidance and are grateful for the opportunity to enhance our manuscript.

Point 3: Introduction: The introduction emphasizes the importance of dynamic variations in endogenous peptides in Chinese human milk across lactation. However, it does not justify why variations might occur across the seven cities studied. It would be useful to clarify whether this was a nutritional hypothesis, as later discussed in the results and discussion, or whether the study should be framed as an exploratory analysis. This section should be expanded to present the scientific rationale more clearly.

The stated objectives also require revision. Given that seven cities and 200 samples are included, proportional sampling representative of each city’s population should ideally have been conducted, and the sample size might need to be larger considering the populations of these cities. The objectives should be revised and clearly stated at the end of the introduction.

Response 3: Thank you for your valuable feedback regarding the Introduction section of our manuscript. Below is our detailed response:

(1) In the revised manuscript, we have expanded the Introduction to clarify the scientific rationale for selecting seven cities, highlighting dietary and lifestyle differences that may contribute to variations in milk peptides. Specifically, we highlighted that China spans a vast geographic area, with substantial differences in dietary culture and lifestyle across regions. For instance, staple food preferences vary between northern cities (e.g., Beijing and Lanzhou) and southern cities (e.g., Chengdu, Jinhua, and Guangzhou). Likewise, coastal cities (Weihai, Jinhua, and Guangzhou) are characterized by a higher consumption of seafood products compared with inland cities (e.g., Zhengzhou, Chengdu, and Lanzhou).

Based on these considerations, this study was designed as an exploratory cross-sectional analysis and analyzed differences and correlations between endogenous peptides and maternal dietary patterns in human milk samples across different cities. We have revised the Introduction accordingly (Lines 65-75).

(2) We have also clarified that this is an exploratory cross-sectional study, rather than a fully representative survey of all Chinese regions. The study objectives have been revised and are now clearly stated at the end of the Introduction (Lines 76-87): as follows: (1) compare endogenous peptide profiles in human milk collected at 2 and 6 months postpartum from mothers across seven Chinese cities representing diverse geographical regions (n=200); (2) compare peptide composition among seven cities in China; (3) investigate potential correlations between peptide profiles and maternal dietary intake to explore potential dietary effects. These findings aim to provide a reference dataset for understanding potential lactation stages and geographic influences on milk peptides, which may inform breastfeeding guidance and infant formula design.

We sincerely appreciate your comments emphasizing the importance of clearly defining research objectives and scientific significance. Thank you.

Point 4: Materials and Methods: The study design should be specified. It is also important to state what type of sampling was performed and whether a sample size calculation was conducted to ensure representativeness for each city. The inclusion criteria for participating mothers should be described in detail to confirm comparability across groups. In particular, it should be clarified whether all participating mothers were exclusively breastfeeding, or if any were supplementing with formula, as this could significantly influence the results.

In the statistical analysis section, it should be indicated how normality of the data distribution was assessed.

Response 4: Thank you for your thorough review of our manuscript and for your insightful comments. We have carefully revised the Materials and Methods section to address these important points. Our detailed responses are as follows:

  • Study design and sampling strategy

We appreciate your request for clarification. The present study is part of the Chinese Human Milk Project (CHMP, NCT03675204), which is a multicenter, cross-sectional study designed to evaluate human milk composition in the Chinese population. The CHMP recruited 1,800 healthy lactating mothers with term delivery from seven Chinese cities: Beijing, Chengdu (Sichuan), Guangzhou (Guangdong), Jinhua (Zhejiang), Lanzhou (Gansu), Weihai (Shandong), and Zhengzhou (Henan). Based on the principle of random sampling, 200 participants were selected from the CHMP cohort for the present analysis. These details have been added to the revised Materials and Methods section (Lines 91-102).

  • Inclusion and exclusion criteria:

We have supplemented the manuscript with the following inclusion and exclusion criteria to ensure comparability across groups:

Inclusion criteria: Women aged 20-40 years, between 15 and 180 days postpartum, who are currently breastfeeding their infants. Self-reported good health. Non-smoker and non-drinker (Lines 105-109).

Exclusion Criteria: Currently undergoing treatment for gastrointestinal symptoms. Suffering from mastitis. Infectious diseases (tuberculosis, viral hepatitis, or HIV infection). Cardiovascular diseases. Metabolic diseases (e.g., diabetes). Psychiatric or neurological disorders. Cancer or other malignant wasting diseases. Recently taken antibiotics (required in some cities). Inability to respond to study questions. Currently participating in any nutritional or pharmacological intervention study.

  • Breastfeeding status

Information on infant feeding practices was collected through questionnaires. Among participants, 70 reported exclusive breastfeeding, 28 reported mixed feeding, and 101 did not provide feeding information. As more than half of the mothers did not report feeding type, this is acknowledged as a limitation of our study. We have now clarified this in the revised manuscript (Lines 105-109) and noted that future research will pay closer attention to differences in peptide composition between exclusive and mixed feeding groups.

(4) Statistical analysis and assessment of normality

Thank you for your thorough review of our manuscript and for your insightful comments. The normality of the data distribution was assessed by IBM SPSS Statistics 26 (USA). As peptide intensity data did not follow a normal distribution, non-parametric statistical tests were applied. Pairwise lactation stage differences (2 vs 6 months) were assessed with Mann-Whitney tests. Regional variations among seven cities were evaluated through Kruskal-Wallis and Wilcoxon tests, applying Bonferroni correction for multiple testing (p < 0.05). These details have been shown in Section 2.6 Data analysis (Lines 172-176).

Point 5: Results and Discussion: At line 172, peptide concentrations are reported at the second and sixth months, but it is not stated whether the differences are statistically significant. The same issue arises in lines 180–181. Even if this information is shown in the figures, statistical significance should be explicitly mentioned in the text. Figure 3 contains valuable information but is difficult to read due to low resolution; the quality should be improved.

The discussion would be strengthened by explicitly addressing the strengths and limitations of the study, as well as by outlining directions for future research.

Response 5: Thank you for your thorough review of our manuscript and your valuable insights. We have carefully revised the manuscript according to your comments. Our detailed responses are as follows:

(1) We have revised the Results section to explicitly indicate statistical significance for the comparisons between peptide concentrations at 2 and 6 months (p> 0.05, Lines 202-203).

(2) Regarding the comparison in Lines 180-181, this section refers to the total number of peptide species identified at 2 and 6 months. Since this analysis represents a descriptive comparison of peptide counts rather than quantitative concentrations, statistical significance testing was not applicable. We have clarified this in the revised text to avoid potential ambiguity (Lines 211-213).

(3) Figure 3 has been replaced with a higher-resolution version to improve readability and ensure that the data can be clearly interpreted (Lines 295).

(4) Furthermore, the Discussion section has been expanded to include a clear statement of the study’s strengths, limitations, and directions for future research (Lines 415-427).

We sincerely appreciate your insightful feedback, which has helped us strengthen the clarity and rigor of our manuscript.

Point 6: Conclusion: The conclusion should be revised in light of the representativeness of the sample studied, to ensure that the claims are proportional to the data presented.

Response 6: Thank you for your insightful comments and for highlighting the Conclusion in our manuscript that require further clarification and improvement. We agree with your comment. The Conclusion has been revised to ensure that the claims are proportional to the data presented and added text: " While these findings provide valuable preliminary insights, the relatively limited sample size and restricted geographic coverage mean that the results should be interpreted with caution. Future research involving larger, longitudinal, and multi-omics cohort studies will be essential to validate these associations and to clarify the functional significance of endogenous peptides in infant development." (Lines 429-448)

Reviewer 2 Report

Comments and Suggestions for Authors

The manuscript is well-documented. It's important to emphasize the novelty of the research compared to previous publications.

My remarks and comments are as follows:

2.4. Mass Spectrometry Analysis and 3. Results and Discussion,

  1. The paper lacks examples of analyses using mass spectrometry. The authors should present sample results in the Results and Discussion section.

Other examples for various sites could be presented as supplementary material.

  1. The authors should also describe the matrix effect.

Did the authors determine the matrix effect?

  1. If the matrix effect was determined, the authors should also add references to the impact factor calculation.
  2. How was the results validated?
  3. Conclusions

The innovative aspect of the research should be more emphasized in the conclusions.

The paper is worthy of publication, once revised and supplemented.

My recommendation for the current version of manuscript: major revision.

Author Response

Point 1: 2.4. Mass Spectrometry Analysis and 3. Results and Discussion,

The paper lacks examples of analyses using mass spectrometry. The authors should present sample results in the Results and Discussion section.

Other examples for various sites could be presented as supplementary material.

Response 1: We sincerely appreciate your valuable comments. The issues you raised are of great importance to us. In the revised manuscript, representative MS/MS spectra of key peptides (e.g., CASB_16–33 peptide) have been incorporated into the Results and Discussion section (Figure 1) to illustrate the analytical process more clearly (Lines 151-154). In addition, supplementary figures (Supplementary Figures S1–S3) now include further MS/MS spectra from milk samples collected across different lactation stages and cities (e.g., Lanzhou and Jinhua). These additions enhance the transparency of peptide identification and provide readers with greater confidence in the robustness and reproducibility of our findings.

We are very grateful to you for the comments that made us aware of the shortcomings in this manuscript. Thank you for your valuable comments.

Point 2: The authors should also describe the matrix effect.

Did the authors determine the matrix effect?

Response 2: We thank the reviewer for raising these critical points regarding matrix effects, which are indeed crucial for LC-MS/MS-based quantitative peptidomic analyses. Although a formal quantitative evaluation of matrix effects was not performed in this study, we employed several well-established sample preparation and quality control measures to mitigate potential matrix interference and enhance the overall reliability of our data.

Specifically, we minimized potential matrix interference by lyophilizing milk samples and reconstituting them in organic solvent prior to analysis. Moreover, a quality control (QC) standard (293 T cell line lysate) was used to ensure analytical accuracy and reproducibility (coefficient of variation < 5% for peak intensity). These methodological details have been explicitly added to Section 2.4 Mass Spectrometry Analysis (Lines 147-149) of the revised manuscript.

This strategy is consistent with methodologies adopted in recent peptidomic investigations (Han et al., 2024; Dekker et al., 2022; Dallas et al., 2015), and provides a practical and robust means to ensure that the instrument response remained stable.

We fully acknowledge that a thorough assessment of matrix effects is imperative for precise quantification. Nevertheless, it is noteworthy that rigorous matrix effect evaluation remains uncommon in the current landscape of human milk peptidomics research. We agree that this represents an important limitation and area for future methodological refinement. We sincerely thank you for this valuable insight.

Point 3: If the matrix effect was determined, the authors should also add references to the impact factor calculation.

Response 3: Thank you for raising this important point. As a formal matrix effect determination was not performed, a calculation of an impact factor was not applicable.

We fully acknowledge that comprehensive assessment of matrix effects is a critical component of rigorous quantitative method validation. However, this remains a major methodological challenge in untargeted global peptidomics due to the vast diversity of unanticipated peptides and the general lack of corresponding isotope-labeled standards for each individual peptide.

The primary objective of the current study was to conduct large-scale exploratory profiling of the endogenous peptidome to identify predominant peptides and characterize trends across lactation stages and geographic regions. We strongly agree with your suggestion, and indeed, we plan to extend this work in future targeted investigations focusing on the most promising bioactive peptides identified herein. This follow-up work will involve the synthesis of stable isotope-labeled internal standards to enable absolute quantification, along with full validation of matrix effects, recovery, and other analytical parameters for those specific peptides.

Thank you again for your critical assessment, which has significantly improved the clarity and rigor of our methodological description.

Point 4: How was the results validated?

Response 4: Thank you for your thorough review of our manuscript and for your insightful comments. We employed two strategies to ensure the reliability and accuracy of our peptide data.

Firstly, to control for instrumental variability and ensure quantitative reproducibility throughout the long sequence of analyses, we implemented a rigorous QC protocol (293 T cell line lysate). And the coefficient of variation of peak intensity was below 5%. This stability is a fundamental prerequisite for ensuring that the quantitative comparisons we make between different human milk samples are technically sound and reliable.

Secondly, to ensure the high confidence of peptide identifications, we applied stringent filters. Only peptides with a high confidence (q-value < 0.01) were included in the results (FDR < 1%).

This clarification has been added to Section 2.4 Mass Spectrometry Analysis (Lines 147-149 and Lines 157-160). Thank you again for your valuable comments.

Point 5: Conclusions

The innovative aspect of the research should be more emphasized in the conclusions.

The paper is worthy of publication, once revised and supplemented.

My recommendation for the current version of manuscript: major revision.

Response 5: We agree with you and have revised the Conclusion to more explicitly highlight the innovative aspects of our study.

Specifically, the study was the first to investigate the characteristics of endogenous peptides in human milk among different lactations and regions from Chinese women, and explore the correlation with the maternal diet (Lines 429-431). These findings could provide implications for both breastfeeding recommendations and the development of infant formula. We appreciate your guidance and are grateful for the opportunity to refine our work accordingly.

Round 2

Reviewer 2 Report

Comments and Suggestions for Authors

Dear Authors,

When planning experiments in the future, Authors should determine the effect of the matrix. My recommendation - Accept in present form.